# The Promise of Behavioral Tracking Systems for Advancing Primate Animal Welfare

**DOI:** 10.3390/ani12131648

**Published:** 2022-06-27

**Authors:** Brenna Knaebe, Claudia C. Weiss, Jan Zimmermann, Benjamin Y. Hayden

**Affiliations:** Department of Neuroscience and Center for Magnetic Resonance Research, University of Minnesota, Minneapolis, MN 55455, USA; weiss349@umn.edu (C.C.W.); janz@umn.edu (J.Z.); benhayden@gmail.com (B.Y.H.)

**Keywords:** big data, deep learning, behavioral tracking, rhesus macaque, primates, behavioral imaging

## Abstract

**Simple Summary:**

Computerized tracking systems for primates and other animals are one of the great inventions of the 21st century. These systems have already revolutionized the study of primatology, psychology, neuroscience, and biomedicine. Less discussed is that they also promise to greatly enhance animal welfare. Their potential benefits include identifying and reducing pain, suffering, and distress in captive populations, improving laboratory animal welfare, and applying our understanding of animal behavior to increase the “natural” behaviors in captive and wild populations, especially those under threat. We are optimistic that these changes will greatly increase the welfare of primates, including those in laboratories, zoos, primate centers, and in the wild.

**Abstract:**

Recent years have witnessed major advances in the ability of computerized systems to track the positions of animals as they move through large and unconstrained environments. These systems have so far been a great boon in the fields of primatology, psychology, neuroscience, and biomedicine. Here, we discuss the promise of these technologies for animal welfare. Their potential benefits include identifying and reducing pain, suffering, and distress in captive populations, improving laboratory animal welfare within the context of the three Rs of animal research (reduction, refinement, and replacement), and applying our understanding of animal behavior to increase the “natural” behaviors in captive and wild populations facing human impact challenges. We note that these benefits are often incidental to the designed purpose of these tracking systems, a reflection of the fact that animal welfare is not inimical to research progress, but instead, that the aligned interests between basic research and welfare hold great promise for improvements to animal well-being.

## 1. Introduction

Non-human primates—monkeys and non-human apes—are found in zoos and research settings in large numbers, and are a crucial animal model in the biomedical sciences [1,2,3,4,5,6]. Captive non-human primates are uniquely susceptible to welfare challenges, due to their social nature, their high level of intelligence, and the fact that they are not domesticated animals [7]. Wild populations also face their own set of welfare challenges, caused by the encroachment of humans. These include problems caused by habitat loss, bushmeat hunting, the illegal pet trade, climate change, and anthroponotic diseases [8]. Because of these welfare challenges, there is a growing ongoing interest in improving welfare for non-human primates. The goal of improving welfare is primarily a moral concern. However, secondary motivations include the benefits that improved welfare offers for enhancing the rigor and reliability of resulting scientific outcomes [9].

Recent years have witnessed great advances in technology that allows for the tracking of animals, including primates [10,11,12]. We use the term “behavioral imaging” to refer to these technologies [13]. Behavioral imaging has attracted a great deal of interest for its potential applications to neuroscience, psychology, zoology, and ethology, among other fields. However, the potential benefits of these technologies for primate welfare have not been well explored. Here we propose that, in addition to its other benefits, the widespread adoption of behavioral imaging will have salutary effects on animal welfare. Specifically, in the current paper we present three main domains where behavioral imaging can improve the welfare of non-human primates: (1) identifying and reducing pain, suffering, and distress in captive populations; (2) improving laboratory animal welfare within the context of the three Rs of animal research (reduction, refinement, and replacement); and (3) using applied animal behavior to promote “natural” behaviors in captive and wild populations facing human impact challenges.

### What Is Behavioral Imaging?

In the past, those interested in understanding the behavior of primates in detail had one option—describing in words what they could see with their own eyes. Technological advances have changed this. Now, we can use digital video cameras with software to monitor the positions of animals’ bodies in space (“pose tracking”). Sophisticated analysis systems can then process this information about pose to determine the behavior the primate was performing (walking, climbing, grooming, etc.) at every moment (“behavioral identification”) in a continuous fashion. We use the term “behavioral imaging” to refer to both of these methods together. Here, as a prelude to exploring their applications to welfare, we briefly review these methods. For a longer review on behavioral imaging in primates, see Hayden et al. (2021) [13], and for a specialized take, see Bain et al. (2021) [14].

Pose tracking: Modern behavioral imaging relies on several major technologies, especially affordable high-quality digital video cameras, image processing software, storage, and deep-learning techniques for analysis of data. These technologies allow for the tracking of primates with high spatial and temporal resolution, often from multiple vantage points, often for long periods of time. These systems record a digital impression of the scene in front of them and, following extensive training, can identify the movement of humans and other animals in the scene. Fundamentally, these systems are based on detailed, annotated training datasets that give example scenes and pose-annotated animals. It is the need for such training sets, rather than technological inadequacies, which tends to serve as the major barrier to progress in these systems. The resulting systems can follow the positions of a few or several landmarks with high spatial and temporal precision. Multi-camera systems can readily provide depth information (the position of landmarks within the three-dimensional scene), while single-camera systems can typically only provide information about the positions of landmarks in the frame. Some recent advances allow for the estimation of depth information from single-view monitoring (called “lifting”) in some circumstances [15,16,17].

Behavioral identification: The detection of pose (the positions of major body landmarks in the scene) is a precursor to the identification of behavior (the categorical identity of what the animal is doing). Some behaviors are identifiable solely from pose—for example, consider the distinction between walking and sitting. Others require contextual information. For example, a monkey may sit in the same position when it is foraging for insects to eat and when grooming its relatives. As a consequence, it is typically non-trivial to identify behaviors based on even perfect reconstructions of pose. Fortunately, recent research has begun to develop the ability to identify specific behaviors [18,19,20,21,22,23,24,25].

Together, these systems constitute tools for behavioral imaging. The fact that these systems do not require human supervision makes them orders of magnitude less expensive than having humans “in the loop”. Their ability to process large volumes of data makes it possible to detect rare and subtle behaviors, as well as changes in behavior over time. Their ability to combine images from multiple views makes it possible to perform complex imaging that humans cannot. The fact that they can classify behavior using unsupervised methods removes the inherent bias of subjective human annotation of behavior [26]. For these reasons, these techniques have engendered a great deal of optimism for scientists interested in animal behavior and animal welfare [27,28,29].

## 2. Behavioral Imaging Can Help Assess Pain and Distress While Improving Welfare

The first step in the improvement of welfare is to measure it. However, measuring welfare is a surprisingly difficult problem [30]. Any animal welfare guidelines will stress the importance of the reduction in pain and distress, but the literature often does not provide clear indications of how to detect these states [31]. Indeed, it takes a large degree of experience with animals for a veterinarian or caretaker to gain the intuition to detect pain or distress. Even then, the time, manpower, and expertise required to monitor animals for signs of pain or distress can be prohibitively costly.

### 2.1. Current Challenges of Assessing Pain and Distress

Researchers conducting biomedical studies on animals are responsible for reducing potential discomfort and promptly treating any pain that may arise [32]. Nonhuman primates are of particular concern because of their high intelligence and their tendency to hide pain from observers. Macaques are notorious for masking clinical signs of illness and injury, possibly to conceal impairments from social group members and/or due to their evolutionary status as prey animals [33,34].

Pain is challenging to define [35,36,37] but is generally understood as the unpleasant sensory and emotional response of an organism experiencing actual or potential damage to its tissue [30]. Pain is more complicated than an animal’s physiological response to noxious stimuli (“nociception”) because it also describes the negative emotional states in response to nociception. These negative emotions are highly subjective, which exacerbates the difficulty of defining pain in general. Without animals possessing the language to describe pain, it is impossible to fully understand their experience [38]. The current best approaches to infer whether an animal is experiencing pain are by monitoring physiological and/or behavioral measures that may indicate pain.

Physiological responses are not particularly effective measurements of pain, as these responses are often delayed (e.g., cortisol measurements in urine); are potentially invasive and stressful in and of themselves (e.g., sampling blood or measuring neurological signatures of nociception via fMRI) [30]; and single measures can indicate either negative or positive states (e.g. changes in heart rate and cortisol/corticosterone levels from emotional arousal can indicate either fear or pleasure) [39]. Poor health responses, while vital to animal welfare, are more likely to indicate extreme issues with welfare rather than more subtle issues; while pathological indicators of distress such as disease, self-injurious behaviors, or death will become apparent given enough time, it is unethical to wait for these outcomes to appear.

Because of the issues associated with physiological detectors, behavioral observations are the superior tool in gaining insight into an animal’s state of welfare, especially in identifying early signs of negative states. There are known indicators of pain in nonhuman primates, such as impaired locomotion, favoring limbs, over-grooming, etc. However, because of their tendency to mask discomfort, nonhuman primates have a strong “observer effect” where they hide these indicators in the presence of an active observer. Research has found that nonhuman primates will resist exhibiting signs of pain and illness (such as hunching with head down, lying down, and dropping food from mouth) when an observer is actively attending to them, even when in moderate to severe states of pain [33]. Despite behavioral observation being more feasible and potentially more reliable than physiological measures, the observer effect makes it particularly challenging for researchers and caretakers to detect signs of discomfort in nonhuman primates. Because of the effect of direct observation on displays of pain and distress, the Institute for Laboratory Animal Research (ILAR) recommends that NHPs should be visually assessed from a distance [30,40]. Unfortunately, this approach does not eliminate observer effects and some behaviors may be subtle or may worsen over time such that an indirect observer will not notice. Video recording can replace behavioral assessments, as cameras will not trigger observer effects. However, these recordings must be annotated by a trained observer to determine which behaviors are abnormal for an individual, and annotation will cause a delay in treatment response. Furthermore, a veterinarian or highly trained staff member must be familiar with normal behaviors at both the species and individual level to ensure proper assessment.

As with pain, it can be challenging to recognize stress in animals. Stress is generally defined as any biological response to perceived threats in the animal’s environment [40]. Stress is not always negative; in small doses, stress responses can help an animal run away from predators, promote social bonding, and/or create resilience to future stressors [41,42]. However, when stressors result in negative pathological responses, the state of the animal elevates to one of distress [30]. Distress occurs when a stressor is chronic, severe, and/or a culmination of several threats, and as such, the biological cost of reacting to the stressor may disrupt normal biological processes. This can eventually lead to pathological responses such as behavioral abnormalities, loss of reproduction, or growth abnormalities, to name a few [43].

Currently, one of the best ways to behaviorally measure stress and distress is by observing stereotypies [30,44,45]. Stereotypies are defined as repeated actions performed as a coping response to stress that has no obvious function to the animal [46]. Stereotypies are assumed to originate as coping responses to environment-induced stress, such as a limited ability to perform species-specific natural behaviors or inadequate social and tactile enrichment [45]. The actions are not initially pathological, but eventually become a learned behavior tool used by the animal to cope with stressful situations. Stereotypical behaviors then integrate into the animal’s behavioral repertoire to a degree that disrupts their normal activities. Consequently, stereotypies can serve as an index of distress and can indicate the quality of care provided to captive animals [47].

Recent technological advances have opened new avenues for indirectly measuring animal emotions. A cutting-edge approach developed in recent years has shown advances in indirectly assessing nervous system activity by measuring electromagnetic radiation emitted from bodies using infrared thermography (IRT) [48]. Thus far, audible distress cues from conspecifics have shown decreases in nasal skin temperature in rhesus macaques and chimpanzees [49,50] and field settings have provided similar results [51]. While this technology is promising, similar physiological markers can indicate different emotional states (e.g., excitement and anxiety). More research must be conducted before IRT can be applied to assess an array of emotional states. Noninvasive eye-tracking technology is another advancement that can be applied to behavioral measurements, as pupil dilation can indicate emotional arousal [52]. However, more research is needed to link eye movements with reliable physiological or emotional health markers [53].

Behavioral measurements of well-being are complicated by their subjective nature and the variability in baseline behaviors between animals. Genetic susceptibility, age, and physiological state of an animal are just a few aspects that influence a behavioral stress response and its biological cost to the individual [30]. For example, an animal who is generally at a higher baseline state of arousal may display more obvious responses to an environmental stressor as opposed to an animal with lower levels of arousal, or vice versa. Therefore, the observer not only must be familiar with indicators of distress specific to the species in question, but also must have an established understanding of the personalities of each animal in a colony to know which behaviors to flag as concerning [54]. Furthermore, certain behaviors (and specifically stereotypies) may be a result of poor living conditions in the past and are therefore not always reflective of current welfare status [55].

### 2.2. Behavioral Imaging Can Assess Pain and Distress

Despite the challenges outlined above, there are indeed signs of poor welfare that can be measured in behavior. We can generally surmise that an animal is in a negative state of welfare if drastic changes in its typical behavior are observed [40]. Examples of these changes may include increased or decreased vocalizations, developing stereotyped or self-injurious behaviors, and/or a change in temperament [56]. These changes may be subtle, and—critically—no single behavior indicates a definitive change. Moreover, these behavioral changes are only meaningful when compared to the animal’s behavioral baseline. As a result, the identification of a large and longitudinal database of the individual animal’s behavior must be part of the assessment process. Pose estimation software provides an excellent means to combine the individualization of each animal observed with reduced cost and manpower, as well as improving reliability. Behavioral imaging systems also allow for continuous observations, including time outside normal working hours or when staff are unavailable. Constant observation allows more data to be collected on each animal and presents the opportunity to alert caretakers of immediate threats to the animal’s well-being on short notice. Ultimately, the most important factor may be one of cost—using humans to assess pain and distress is feasible, if imperfect, but requires expensive, highly-trained observers.

While behavioral imaging can solve manpower and expense issues, an added benefit is that behavioral imaging removes the limitations and biases of humans. By providing a standardized method of assessment, this software removes inter-observer unreliability. Additionally, since human perception is biased by cognitive processes, automatic tracking systems can remove the tendency to anthropomorphize animal behaviors [57,58]. Behavioral imaging also has the potential to detect gradual changes in the nature and frequencies of behaviors, as well as subtle signs of pain or discomfort that could go unnoticed by human observers. For example, by assessing abnormal behaviors against the baseline of a specific animal, this software could determine that grooming frequency or intensity has changed over time, allowing caretakers to intervene by removing apparent stressors or by providing enrichment before obvious signs of alopecia occur. Behavioral imaging software could potentially be sensitive enough to detect very subtle behaviors that a trained human observer may not notice, such as a disguised limp. Indeed, a behavioral imaging system could even detect second-order changes in behavior, such as a reduction or delay in walking behavior, which reflect the animal’s attempts to hide its pain.

Automated pose tracking systems can also expand our knowledge of animal welfare because they can potentially detect welfare indicators that were previously unknown. This would require novel research—in particular, it would require ground truth data that have been validated and then used to train the system. Once this is completed and if it is replicated, known linkages could be used in future welfare efforts. These systems can have a much broader field of view than human perception. For example, deep-learning systems can detect behaviors that are too fast or too slow for human observers to detect, that involve multiple small signals, that have high structural and temporal complexity [59], that are too subtle for humans, or that humans may not recognize. Furthermore, automated pose-tracking systems can be used in conjunction with new imaging-based technologies such as IRT and restraint-free eye-tracking to enhance our understanding of the connection between behavior, physiology, and emotional states.

While the issue of pain and distress is perhaps most relevant to the confined spaces and testing requirements of laboratory environments, the same principles hold true in other captive populations, including zoos, wildlife rescues, and field cages. If specialized software can learn to detect pain and distress in lab animals, it has the potential to improve the lives of the animals in any captive setting.

### 2.3. Behavioral Imaging Can Improve Welfare

Behavioral imaging has a clear role in identifying pain and distress in animals, but it also has the potential to help actively improve welfare by creating more enriching environments. Historically, attempts to improve the welfare of zoo or laboratory-housed animals have focused on environmental “inputs” (e.g., cage/enclosure size, the addition of enrichment, allowing for social interactions), with less focus on evaluating the animals’ “output” (i.e., their physiological, health, or behavioral response to these environmental inputs) [60,61,62]. A focus on environment over physiology, health, or behavior may partially be caused by the challenges of measuring these responses (as outlined above). However, despite these challenges, a focus on animal outputs is crucial because efforts to improve welfare based on environmental input may be directed ineffectually or may potentially cause more harm than good. For example, in response to the small enclosure sizes of early zoos, new designs focused on increased space. However, studies have found that quality of space appears to outweigh quantity of space for great apes [63] and social factors may outweigh spatial factors for macaques (and presumably for several other primate species as well) [64].

Fortunately, indicators of both optimal and poor welfare manifest as specific behaviors. For example, interventions should increase normal behavior while decreasing self-injurious or otherwise negative behaviors [40]: social companions should lead to an increase in affiliative behaviors rather than be a cause for avoidance and stress; enrichment should encourage exploration and species-typical foraging behaviors rather than produce neophobic responses, conflict with conspecifics, or excessive foraging at the expense of other healthy behaviors. There is a clear need for objective and sensitive measures of behavior, and behavioral imaging provides a new resource to fulfill these aims.

## 3. Behavioral Imaging Can Improve Welfare within the Context of the Three Rs of Animal Research

The three Rs of animal research aim to improve animal welfare by Replacement of animals with alternative models, Reduction in the number of animals used for a study, and Refinement of study methods and housing/husbandry to minimize pain, suffering, and distress [65]. As outlined earlier, research on nonhuman primates is both unavoidable and fraught with welfare challenges [7]. The three Rs therefore represent specific goals than can move us in the direction of improving welfare. Behavioral imaging can contribute to each of the three Rs.

In this section, we focus on the benefits of behavioral tracking to neuroscience in particular. We choose neuroscience as an example domain where imaging can contribute to the three Rs partly because of its importance and visibility in primate science, and partly because of our personal interest in the field. Broadly speaking, rhesus macaques (and sometimes Japanese macaques) are often used as a model organism for human brain activity. (As are, increasingly, marmosets; [66,67,68,69]).

### 3.1. Replacement

The principle of Replacement holds that we can in some cases improve animal welfare by replacing research animals with animal-free approaches (such as computer models or in vitro methods). Classically, neuroscience experiments are bespoke—which means that we devise a hypothesis and design the simplest possible experiment to test it. This is changing. We are entering into the era of Big Data in neuroscience [70]. This means that we can collect data of much higher quantity than in the past. In practice, big data in neuroscience come from collecting hundreds or even thousands of neurons at a time, collecting whole-brain high-resolution scans at high field strength, or other methods that provide orders of magnitude more data than traditional bespoke methods. These Big Data methods allow for the collection of so much data that they require new analysis techniques, some of which may not have been invented yet [71]. Big Data sets also provide the opportunity for post hoc experimentation—that is, doing experiments on data that already exist in databases. This is important because such experimentation can greatly extend the utility of experiments, meaning many more results can come from a dataset than in the past, and that in turn results in Reduction (see Section 3.2 below). For example, our own lab has managed to make use of a single carefully designed gambling task to test several hypotheses that were not anticipated at the time of data collection [57,72,73,74,75,76,77,78,79,80,81,82,83,84,85]. These papers represent important advances, some of which make use of statistical techniques that did not exist when the data were collected or were motivated by novel hypotheses that did not exist at the time the data were collected [86,87,88], but would have required the collection of new data to test. Consequently, they represent an area where big data replaced novel experimentation, and thus contribute to the goal of Replacement.

The major limitation in all of this work, however, is that the understanding of neural data is usually best performed in conjunction with behavior. While neural data can now be *big*, behavior, as it is normally measured, cannot. Behavioral imaging can resolve this issue and make behavioral datasets as rich and fertile for post hoc research as neural datasets. Without behavioral imaging, the behavior will act as a bottleneck—that is, extremely detailed brain data with very simple behavior can only go so far in helping to test neuroscientific hypotheses. On the other hand, the extremely rich data generated by behavioral imaging, when registered with detailed neuronal data, promises to create datasets large enough that they can lead to experiments being conducted in silico for years.

### 3.2. Reduction

Reduction refers to the goal of using fewer research animals. Behavioral imaging can also contribute to Reduction, or the design of experiments that use non-human primates in smaller numbers. First, simply by providing more behavior, the need for more animals is correspondingly reduced. That is, the ability to test scientific hypotheses requires a certain amount of data to overcome uncertainties associated with noise. Sometimes, additional data need to be from independent samples, and in these cases, reduction is impossible (but see below). However, in many cases, the independence of data is not a limiting factor. In these cases, one can obtain more data with fewer resources. To choose another example from our own work, our lab has begun to study the neural basis of pursuit decisions, in which macaques use a joystick to pursue a fleeing prey in a virtual (computer screen) environment [85,89]. While these tasks do not involve behavioral imaging, they involve continuous 3D movement, and we can readily track this behavior using conventional computer systems. This rich behavior has allowed us to derive complex behavioral models that can address important outstanding questions, with a smaller amount of data that would have been required to address these questions using conventional binary choice tasks. In other words, the use of a continuous task design with measurement allowed us to meet the goal of Reduction with no sacrifice in the validity of the data for our hypotheses.

Relatedly, by providing more and richer behavior for each neuron collected, the scientific value of each neuron is enhanced. As a result, the quality of inferences that can be drawn from each neuron is enhanced. Therefore, the scientific community can obtain the same results with fewer research animals.

Third, by providing more naturalistic behavior, the validity of the behavior that is collected is improved. Typically, experiments are performed using unnatural but simple tasks that recapitulate important behaviors, but these tasks may lack external validity. Using more naturalistic tasks, ones with continuous motion and many small decisions, can make the behavior more relevant to answering questions [90,91]. These basic features mean that the number of animals needed to make a given discovery can be reduced. It is not clear yet, because the field is young, how much reduction is possible, but we are sanguine.

To be a bit more speculative, the promise for reduction potentially may be even greater. We suspect that there are some important hypotheses that could be answered with brain measures and not with conventional behavioral measures, but that can be answered with high-quality behavioral imaging and without brain activity. For example, consider that one may use activity in some brain region as an index of some inferred variable. A typical case would be the value of an option to a subject—one may want to know how much the subject values the object and may not, for some reason, trust the decision-maker’s self-report. This may be true in cases of studying deception, in studying competitive games, in studying subjects who are not incentivized to report accurately, or in studying cases where subjects have difficulty explaining the value of their options. However, behavior may be a tell—for example, the vigor with which a subject moves may indicate their subjective value [92]. If the behavior is detailed enough and if enough of the behavior is available to the imaging system, it may be possible to entirely replace the neuronal measure with a behavioral one. This is not to say that imaging will replace invasive measures completely, but in some specific cases, it may do precisely that. In other words, the amount of information about the subject’s internal state revealed by behavior may be so great that it obviates the need for invasive measurements.

### 3.3. Refinement

Refinement refers to the goal of using techniques that are less invasive or otherwise have negative welfare consequences. NHP behavioral studies commonly require the animal to be placed in a primate chair or have its movement restricted in some other way. While restricting the movement of subjects has practical benefits, there are several scientists beginning to experiment with less restrictive data collection methods, for example cage-based cognitive testing in animal housing environments such as touchscreen “kiosk” stations [93]. Just to give one example, we have developed touchscreen kiosks for our monkeys, and allow them to perform gambling tasks; we find that the data generated in these contexts is just as good (indeed, may be more externally valid) than data generated using conventional chaired techniques [94]. While these kiosks have scientific benefits, such as improved ecological validity (because tasks allow for unrestrained species-typical behaviors) and reduced manpower (because the animal does not need to be handled), they also improve welfare by allowing for more free movement in the world, which animals typically find rewarding.

Indeed, in-cage touchscreens have been shown to be a form of cognitive enrichment and also allow for autonomy, provision of choice (e.g., the choice of which task to engage in, at what time, and for how long), and a sense of agency, which are imperative for psychological well-being [94,95,96]. Handling and transfers to primate chairs are also a source of stress eliminated by home-cage kiosks. Touchscreen kiosks also may allow for a Reduction in the number of animals used in the study as a single animal, with access to a kiosk all day, can work multiple times per day, when it is most motivated (an animal’s time preferences may change daily and be difficult to predict by investigators). Despite these benefits to research and welfare, there remain unresolved issues with kiosks that may prevent their widespread use. As the animal is housed in a colony room, there will invariably be distractions from conspecifics and husbandry teams, potentially impacting the quality of the data. It is also difficult to determine if the animal is fully engaged in the task (other than trials completed); they may interact more with the task throughout the day because it is available but may be less focused, motivated, or enthusiastic.

Behavioral imaging has the potential to resolve these issues by determining whether trials are completed while the animal is focused and are therefore valid or whether, through objective means, data should be discarded. This software can also assist with training; for example, animals can be monitored and rewarded for being calm and focused, and a closed-loop system, whereby if an animal deviates from the task or loses engagement there is mitigation or an intervention for re-engagement, can produce cleaner data, refined training, and reduced stress in the animal. Behavioral imaging also has the potential to provide a better understanding of how the animal performs the task. Taken together, by reducing the drawbacks associated with kiosks, behavioral imaging can improve welfare by allowing for increased use of these beneficial devices.

## 4. Behavioral Imaging Can Improve Our knowledge of “Natural” Behavior and This Knowledge Can Be Applied to Improve Welfare in Captive and Wild Population

The field of applied animal behavior focuses on applying our understanding of animal behavior to improve the welfare of captive and wild animals [97]. It often uses the behavior of wild animals to inform this purpose. This goal necessitates both a clear understanding of the frequency and types of behaviors displayed by wild animals in their natural habitats and the differences between these behaviors and those displayed by their captive counterparts, newly released animals, or wild animals in disturbed landscapes. For example, if we determine that wild marmosets spend approximately 10% of their day engaging in predator surveillance while a group of captive marmosets or wild marmosets living in disturbed landscapes spend 20% of their day engaged in these behaviors, caretakers could add hiding places or canopy covers and reassess time budgets to ensure that predator surveillance, and likely stress, has decreased. On the other hand, we may be able to rest assured that marmosets who spend 10% of their day on surveillance are not a source of concern.

Thus far, research on the behavior of wild animals relies on either direct observation or camera trapping, both of which require large amounts of manpower to identify and code behaviors, and are fraught with issues such as inter-reliability in coding, required time to train coders, and, in the case of direct observation, disruption to natural behaviors due to the presence of a human observer. As a result, while these methods are valuable in identifying broad behavioral categories (e.g., geophagy, predation events, and food extraction methods), they are less suited to identifying more subtle behaviors and behavioral patterns. The use of behavioral imaging to survey the frequency and types of behaviors displayed by wild animals eliminates many of these issues by providing consistent and unbiased ethograms and can process much larger amounts of data than would be feasible by human coders, in addition to the detection of both gross and nuanced behaviors. While pose estimation software cannot eliminate all the shortcomings of camera trapping, many of these remaining issues, such as limited coverage of large territories and disturbances caused by the camera itself, are less important in the context of captivity. For example, while placing camera traps in fixed locations where specific behaviors or resource use occurs biases the types of behaviors observed, these behaviors are most relevant to captive animals whose enclosures are intended to mimic these important locations.

### 4.1. Captive Animals

Designing habitats that allow animals to engage in natural behaviors is of large importance to zoos and other wildlife centers, for both the health and wellbeing of the animals in addition to more rewarding experiences for zoo patrons. Understanding how animals use resources, engage with their space, and interact with each other socially can provide insight into the requirements of a zoo enclosure and allow for continued assessment and improvement of these spaces. Understanding complex social behaviors is also of utmost importance for animals living in restricted spaces with conspecifics and in the case of captive breeding (which, depending on the species, can be challenging without a deep understanding of natural courtship and mating behaviors) [98]. Pose estimation software can allow for a better understanding of social organization, signaling behavior, and mate choice, which can inform where resources should be dedicated in order to encourage captive breeding and prevent pair or social group breakdown.

### 4.2. In Situ Conservation

With habitat fragmentation and human disturbance ever increasing, it is important to identify the behavioral responses to these threats and to determine the best course of action to improve the welfare of these populations. Applied behavioral research is vital to these animal welfare and conservation efforts, as behavioral mechanisms play a major role in mediating the impacts of habitat fragmentation and human disturbance on vulnerable species [99]. Understanding natural behaviors and behavioral responses to new or altered landscapes can inform reserve and habitat corridor design, determine the success of translocated or newly released animals, and identify whether anthropogenic nuisances (e.g., noise, artificial light, roads and traffic, and ecotourism) are habituated to or result in chronic stress, altered time budgets, or suppressed reproduction [98].

## 5. Conclusions

With non-human primates being such a large focus of laboratory research, zoos, and conservation efforts, there is a clear need to consider the welfare of these highly intelligent, sensitive, and social creatures. While necessary, we currently have poor means to institute such change. Behavioral imaging has the potential to revolutionize welfare management by providing real-time analysis of behavior that can detect signs of pain, distress, or other welfare challenges that would otherwise require unfeasible amounts of manpower and expertise. It can also be used to improve the quality and quantity of data collected in scientific studies, reducing the number of animals required and the pain and stress experienced by those animals, without compromising data integrity. Finally, behavioral imaging can extend beyond a mere reduction in pain and suffering to actively improve the lives of captive and wild primates by guiding interventions that allow animals to express the behaviors they would naturally. We are excited to see what this technology holds for the future.

## Data Availability

Not applicable.

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
