# Peer review of "The Promise of Behavioral Tracking Systems for Advancing Primate Animal Welfare"

_animals, 2022, doi:10.3390/ani12131648_

Round 1

Reviewer 1 Report

· Non-human primates - monkeys and non-human apes is very important animal model in the biomedical sciences. The welfare of non-human primates are more and more concerned. This manuscript is very importanced for monkeys and non-human apes welfare.Recent years have witnessed major advances in the ability of computerized systems to track the positions of animals as they move through large and unconstrained environments. These systems have so far been a great boon in the fields of primatology, psychology, neuroscience, and biomedicine. The manuscript discuss the promise of these technologies for animal welfare. Their potential benefits include identifying and reducing pain, suffering, and distress in captive populations, improving laboratory animal welfare within the context of the three Rs of animal research (reduction, refinement, and replacement), and applying our understanding of animal behavior to increase the “natural” behaviors in captive and wild populations facing human impact challenges. I suggest accepting this manuscript without any modification.

Author Response

We thank the reviewer for their review and for their kind words.

Reviewer 2 Report

The paper is nice but I think that it misses illustrations and examples at different places in order to improve the author’s arguments.

Lines 93-94: you should describe some of the methods/research/software that were developed to identify such poses/behaviours as Deeplabcut or some other deep learning processes.

1.2. What is behavioural imaging? > This part could be a bit more developed with some examples and a figure illustrating how the automatic data collection is done. You can for instance add this citation where face and behaviour identification are made on chimpanzees > Bain, M., Nagrani, A., Schofield, D., Berdugo, S., Bessa, J., Owen, J., ... & Zisserman, A. (2021). Automated audiovisual behaviour recognition in wild primates. Science advances, 7(46), eabi4883.

Line 123: please quickly define pain, with the three components (subjective, behavioural and physiological).

Line 177 and in other places: I think you can also talk about thermography to measure stress and eye-tracking to follow specific behaviour of animals

Cutting-edge infrared thermography as a new tool to explore animal emotions, Y Sato, F Kano, S Hirata, Japanese Journal of Animal Psychology, 68.1. 7

The application of non-invasive, restraint-free eye-tracking methods for use with nonhuman primates. LM Hopper, RA Gulli, LH Howard, F Kano, C Krupenye, AM Ryan, ... Behavior Research Methods 53 (3), 1003-1030

Please read this paper about animal agency. I think that you can use many references and this paper to enhance your proposals in your review. Cédric Sueur, Sarah Zanaz, Marie Pelé. What if animal agency could improve behavioral and neuroscience research?. 2022. ⟨hal-03299505v4⟩ https://hal.archives-ouvertes.fr/view/index/docid/3684118

Line 227: complexity of behaviours could also give a cue about stress of animals and we could imagine automatic way to measure it: MacIntosh, A. J., Alados, C. L., & Huffman, M. A. (2011). Fractal analysis of behaviour in a wild primate: behavioural complexity in health and disease. Journal of the Royal Society Interface, 8(63), 1497-1509.

Line 268: Indeed, human perception is biased by cognitive processes and automatic tracking can remove this bias > Pelé, M., Georges, J. Y., Matsuzawa, T., & Sueur, C. (2021). Perceptions of Human-Animal Relationships and Their Impacts on Animal Ethics, Law and Research. Frontiers in Psychology, 3736.

AND Sueur, C., Forin-Wiart, M. A., & Pelé, M. (2020). Are they really trying to save their buddy? the anthropomorphism of animal epimeletic behaviours. Animals, 10(12), 2323.

Line 277: this is a false principle, lower animals do not exist, at least in the European declaration on animal experimentation. All vertebrates are considered similarly in terms of pain. Please refer replacement only with alternative methods not using animals.

For each of the point 3.1, 3.2 and 3.3 could you please give a specific instance of what behavioural imaging could conduct to? What kind of new results could we get?

Point 4. Behavioural tracking could also lead to new research in the field > Please read Cédric Sueur, Sarah Zanaz, Marie Pelé. What if animal agency could improve behavioral and neuroscience research?. 2022. ⟨hal-03299505v4⟩ https://hal.archives-ouvertes.fr/view/index/docid/3684118

Reviewer 3 Report

The review is clear. The application of computerized tracking system can simplify the identification of pain and distres behaviors in primates as weel as in other animals allowing to increase animal welfare. 

Author Response

We thank the reviewer for their review.